# Assessment of Ecosystem Health and Its Key Determinants in the Middle Reaches of the Yangtze River Urban Agglomeration, China

**DOI:** 10.3390/ijerph19020771

**Published:** 2022-01-11

**Authors:** Fengjian Ge, Guiling Tang, Mingxing Zhong, Yi Zhang, Jia Xiao, Jiangfeng Li, Fengyuan Ge

**Affiliations:** 1Department of Land Resource Management, School of Public Administration, China University of Geosciences, Wuhan 430074, China; fjge@cug.edu.cn (F.G.); jfli0524@163.com (J.L.); 2International Education College, China University of Geosciences, Wuhan 430074, China; 3Tourism College, Xinyang Normal University, Xinyang 464000, China; 4School of Economics and Management, China University of Geosciences, Wuhan 430074, China; yizhangcug@163.com; 5Hubei Province Culture Tourism Investment Group Co., Ltd., Wuhan 430074, China; ecotic_xj@163.com; 6The Heart of Forestry Development in Changle County, Weifang 262400, China; cllingaiban@163.com

**Keywords:** spatiotemporal pattern, GeoDetector, sustainable development, ecological environment

## Abstract

Urban agglomerations have gradually formed in different Chinese cities, exerting great pressure on the ecological environment. Ecosystem health is an important index for the evaluation of the sustainable development of cities, but it has rarely been used for urban agglomerations. In this study, the ecosystem health in the middle reaches of the Yangtze River Urban Agglomeration was assessed using the ecosystem vigor, organization, resilience, and services framework at the county scale. A GeoDetector was used to determine the effects of seven factors on ecosystem health. The results show that: (1) The spatial distribution of ecosystem health differs significantly. The ecosystem health in the centers of Wuhan Metropolis, Changsha–Zhuzhou–Xiangtan City Group, and Poyang Lake City Group is significantly lower than in surrounding areas. (2) Temporally, well-level research units improve gradually; research units with relatively weak levels remain relatively stable. (3) The land use degree is the main factor affecting ecosystem health, with interactions between the different factors. The effects of these factors on ecosystem health are enhanced or nonlinear; (4) The effect of the proportion of construction land on ecosystem health increases over time. The layout used in urban land use planning significantly affects ecosystem health.

## 1. Introduction

With the continuous development of the global population and social economy, cities will become an inevitable product of the development of human society [1]. Based on the implementation of reform and opening-up policies, the Chinese economy has rapidly developed, especially in the past 40 years. The urbanization rate reached 59.58% in 2018, which means that more than half of the population lived in a city. The social form was also changed to focus on cities. With the rapid development of human society, various ecological problems have become apparent. National and global ecological and environmental problems, such as energy shortages, water pollution, land degradation, biodiversity loss and the frequent occurrence of extreme weather, have emerged [2,3]. Urban expansion has affected large swathes of cultivated land, forest land and water. Because of the continuous increase in the population, more resources and space are required, which damages the structure and functional stability of ecosystems to varying degrees by exceeding the carrying capacity. Once this happens, the stability, resistance, and resilience of the ecosystem change, reducing the ecosystem’s ability to provide ecosystem services and endangering human health and economic development. Because of severe ecological and environmental problems, studies on ecosystem health have attracted global attention.

The urban ecosystem includes many aspects, such as ecology, social economy, and human health. Its health status and driving factors also attracted increasing attention from scholars [4,5,6,7]. The concept of urban ecosystem health combines the ability to meet the reasonable needs of human society with the ability to maintain self-renewal and self-generation based on previous ecosystem health experience. The most important feature of urban ecosystems is that they are controlled by humans [8]. For example, Hancock [9] explained the concept of urban health by addressing the correlations among the economy, environment, and society. From the perspective of ecology and sociology, Guo et al. [10] pointed out that reasonable structure and function must be maintained in a healthy urban ecosystem, which must be able to provide sustainable ecosystem services to urban residents.

At present, mainly two research methods are used to evaluate ecosystem health: indicator species and index system methods [11,12,13,14]. The indicator species method is mainly used to evaluate aquatic ecosystem health, whereas the index system method is more suitable for the study of composite ecosystems. The index system method provides more detailed information, and thus is more suitable for evaluating urban ecosystem health. Although many models were developed for the evaluation of ecosystem health, there is a lack of unified and mature evaluation index systems and methods worldwide [4,8,14]. The Stress–State–Response (PSR) conceptual model was established by the Organisation for Economic Co-operation and Development and the United Nations Environment Programme. The model was constructed based on an index system used for the evaluation of the landscape pattern of coastal ecosystem health, which is one of the most widely used methods for ecosystem health evaluation [13,14]. Costanza et al. (1992) proposed a method that can be used to calculate ecosystem health by integrating ecosystem health indicators.

Urban ecosystems are centered on human life, and healthy conditions must include human well-being. Based on previous research, ecosystem services should be included in the assessment of urban ecosystem health [15,16]. Ecosystem services are functions of the natural ecosystem and human well-being and are important indicators of ecosystem health [15]. Highlighting these indicators with respect to the general characteristics of the ecosystem is important for the evaluation of urban ecosystems [17]. Therefore, the ecosystem vigor, organization, resilience, and services (VORS) framework was established for the assessment of urban ecosystem health based on the classic index system [18]. This framework is based on the ecosystem structure, function, process, and service research paradigm, and focuses on the combination of natural ecosystem health and human health [19,20].

Many studies evaluated urban ecosystem health in China in first-tier cities, such as Beijing [21], Shanghai [22], Guangzhou [23], and Shenzhen [15], as well as Zhuhai [13], Qiqihar [24], Baotou [25], and other small- and medium-sized cities. In addition, comparative studies were carried out in important Chinese cities [26]. The future development of the entire metropolitan area and urban agglomeration also affects the surrounding ecosystem [27]. However, ecosystem health in urban agglomerations has been evaluated in few studies [10,20]. In the present study, the middle reaches of the Yangtze River Urban Agglomeration (MRYRUA) were used for a case study to explore the spatiotemporal evolution of ecosystem health and the factors controlling it at the county scale. The MRYRUA is a large-scale, national-level urban agglomeration including the Wuhan Metropolis, Changsha–Zhuzhou–Xiangtan City Group, and Poyang Lake City Group, which play an important role in China’s regional development pattern. However, many environmental problems arise during the development of urban agglomerations. The material life, spiritual civilization, and urban life that people enjoy come at the expense of the natural environment. Especially after industrialization, rapid population growth and the improvement of the ability to obtain natural resources caused resource shortages, environmental pollution, a sharp reduction in species populations and excessive land use. Cities have gradually become unhealthy and undesirable, and are termed “urban diseases”. Therefore, the spatiotemporal evolution of ecosystem health in the MRYRUA was explored in this study to provide a scientific basis for decision makers in sustainable development. The objectives were to: (1) analyze the spatiotemporal evolution of ecosystem health in the MRYRUA and (2) explore the factors controlling ecosystem health changes in the MRYRUA.

## 2. Materials and Methods

### 2.1. Study Area

The MRYRUA extends throughout a large area of China (Figure 1). The whole area is undergoing the same rapid urbanization process and is an important part of the Yangtze River Economic Belt. It is also a key area for the implementation of the strategy for the rise of the central region, realization of reform and opening-up policies in all directions, and the promotion of new urbanization. It occupies an important position in regional development patterns. In 2015, the National Development and Reform Commission issued the Development Plan for the Middle Yangtze City Cluster, which made the MRYRUA the new economic growth pole of China, a pioneering area for new urbanization in the Midwest, an inland open-cooperation demonstration zone, and man-made area of “two-type” social construction. In 2018, the Central Committee of the Communist Party of China and the State Council requested Wuhan to be the center of the development of the MRYRUA.

### 2.2. Data Sources and Processing

In this study, net primary productivity (NPP) data (1 km resolution) were obtained from the National Science and Technology Basic Condition Platform National Earth System Science Data Center (Available online: http://www.geodata.cn (accessed on 20 April 2021)). The digital elevation model (DEM, 30 m resolution) was obtained from the Geospatial Data Cloud (http://www.gscloud.cn/) (accessed on 12 March 2021). The mean annual precipitation, normalized difference vegetation index (NDVI, 1 km resolution), and land use data (30 m resolution) were derived from the Landsat satellite 30 m land use raster data product from the Resource and Environmental Science Data Center of the Chinese Academy of Sciences (http://www.resdc.cn/Default.aspx) (accessed on 20 April 2021). These data included six first-level land types, including cultivated land, forest land, grassland, water area, construction land, and unused land, and 25 second-level land types such as paddy fields, dry land, and forest land. Landscape indices, such as Shannon’s diversity index (SHDI), are based on original land use data and were obtained using Fragstats 4.2 software. Population data (100 m resolution) were obtained from worldpop (https://www.worldpop.org/) (accessed on 15 April 2021). 

### 2.3. Methods

#### 2.3.1. Ecosystem Health Assessment Framework

Based on the VORS framework, an ecosystem health index was constructed for the MRYRUA [15,16]. Because each factor is key and indispensable in assessing ecosystem health, multiplication was used. Considering that the dataset after normalization was too small, and the differences among the data may be weakened, the root of the fourth power was applied:(1)$$EHI=\sqrt[{4}]{V\times O\times R\times S},$$
where *EHI* is the ecosystem health index and *V*, *O*, *R*, and *S* are the ecosystem vigor, organization, resilience, and services, respectively. Referring to the results of previous research, the average ecosystem health of four years at the ordinary level was defined as 0.468–0.483, and ecosystem health was divided into the following five categories: well (0.6–1.0), relatively well (0.5–0.6), ordinary (0.4–0.5), relatively weak (0.3–0.4), and weak (0–0.3). We used these five ecosystem health levels to evaluate the spatiotemporal characteristics of ecosystem health in the MRYRUA.

The ecosystem vigor characterizes the function of the ecosystem and describes the metabolism or primary productivity of the ecosystem [15]. The NPP, which is the basis of ecosystem functions, is the total net organic matter produced by photosynthesis and the total energy provided by primary producers to other components of the ecosystem. The NPP is an effective indicator of the effect of urban development on regional logistics and energy flow, and thus can be used to characterize the ecosystem vigor [28]. Ecosystem organization represents the structure of an ecosystem and describes the interactions among various components of the ecosystem. This depends on the landscape heterogeneity (LH) and landscape connectivity (LC). The LH is represented by the landscape diversity, which can be determined with the SHDI [29]. Although the landscape appears to be disordered based on the surface LH, the LH mitigates significant changes of the landscape, leading to a more organized, dynamic, and stable landscape [30]. The LC depends on the overall connectivity of the landscape and that of important ecological patches (e.g., forest land in this study) [31]. In this study, Contagion (CONTAG) and Patch Cohesion Index (COHESION) were used to quantify the overall connectivity of the landscape and that of forest land, respectively. Considering that the LH and LC of the landscape have slightly greater effects on the health of the ecosystem and cannot replace each other, their weights were set as equal (0.35) [15]:(2)$$\begin{array}{l} O=0.25\times SHDI+0.1\times AWMPFD+0.25\times FN1+0.1\times CONT1\\ +0.1\times FN2+0.05\times COHESION1+0.05\times FN3+0.025\\ \times COHESION2+0.05\times FN4+0.025\times COHESION3 \end{array}$$
where *SHDI* and *AWMPFD* represent the *SHDI* and area-weighted average patch fractal index, respectively; *FN*1, *FN*2, *FN*3, and *FN*4 represent the landscape, forest land, water, and wetland fragmentation index, respectively; *COHESION*1, *COHESION*2, and *COHESION*3 represent the forest land, water, and wetland *COHESION*, respectively; and *CONTAG* is the landscape contagion index.

Ecosystem resilience represents the ability of landscape patches to maintain their original function and structure under the interference of natural and human factors, reflecting the resistance and adaptability of landscape patches to external interferences of ecosystems [31]. It is easier to recover a land use type that has characteristics similar to those of the natural ecosystem when subjected to external interference [15]. Therefore, the resilience coefficient (RC) was set according to the difficulty with respect to the recovery of different land use types [15,31] (Table 1) and the same type of internal differences based on the *NDVI*:(3)$$RC_{i}=\frac{NDVI_{i}}{NDVI\_mean_{j}}\times RC_{j}$$
where *RC_i_* represents the resilience coefficient of the *i*th grid, *NDVI_i_* is the *NDVI* value of the *i*th grid, *NDVI_mean_j_* is the average *NDVI* value of class *j* in which the grid *i* is located, and *RCj* represents the resilience coefficient of the land category *j*.

This study mainly considered Xie et al.’s improved coefficient of services value for China’s terrestrial ecosystems, corresponding to secondary land use types with ecosystem secondary classification. The sum of the service values is the coefficient of ecosystem services (SC) [32]. The results of previous studies indicated a good correlation between the NDVI and the value of ecosystem services [33]. Therefore, the NDVI ratio with the shrub forest was calculated for second-level land types that cannot directly correspond, such as sparse forest land and other forest land. Similarly, the medium-cover grassland was calculated based on the correlation between the NDVI ratio and high-cover grassland and the SC of the second-level land type was obtained (Table 1). The index system of urban ecosystem health assessment, based on the VORS framework, is shown in Table 2.

#### 2.3.2. GeoDetector

In this study, the GeoDetector proposed by Wang et al. [34] was used for the correlation analysis of factors affecting ecosystem health in the MRYRUA. The model quantitatively measures the importance of independent variables relative to dependent variables by analyzing the overall differences between various types of geographical locations. It is widely used to understand the social economy and ecological environment [35,36,37]. The calculation of the GeoDetector is as follows:(4)$$q=1-\frac{1}{N\sigma^{2}}\sum_{h=1}^{L}N_{h}\sigma_{h}^{2},$$
where *q* is the index for detecting the factors affecting the spatial differentiation of ecosystem health, *N* is the number of all units in the study area, *N_h_* is the number of sample units in layer *h*, *h* is the classification of factors affecting ecosystem health, *L* is the number of factors, the value range of *h* is [1, L], and *σ^2^_h_* and *σ^2^* are the variance of a certain layer of *h* and that of the whole region, respectively. The range of *q* values is [0, 1]. The larger the *q* value is, the stronger the stratified heterogeneity. A *q* value of 1 indicates that the factor completely controls the spatial distribution of ecosystem health; a *q* value of 0 indicates that the spatial distribution of ecosystem health is not affected by the factor.

The interaction detector was used to determine the effects of the interactions between different factors on the spatiotemporal changes in ecosystem health. A healthy spatial pattern of the urban agglomeration ecosystem in the MRYRUA formed under the combined effects of various factors. In this study, seven representative factors were selected to determine the mechanism responsible for the formation of healthy spatial patterns of urban ecosystems: the proportion of construction land (X1), proportion of forest land (X2), proportion of water (X3), land use degree (X4), population (X5), average annual precipitation (X6), and DEM (X7). The land use degree was calculated by using the land use degree measurement model proposed by Zhuang and Liu [38]; it represents the state of the maintenance of a natural balance in the Chinese land system under the influence of social factors.

## 3. Results

### 3.1. Spatiotemporal Characteristics of Ecosystem Health in the MRYRUA from 2000 to 2015

From the perspective of spatial distribution, the ecosystem health in the central areas of Wuhan Metropolis, Changsha–Zhuzhou–Xiangtan City Group, and Poyang Lake City Group below average. The center of Wuhan Metropolis has been at a weak level for a long time (Figure 2). Units with a well ecosystem health level are relatively few and mainly distributed in areas far away from the central city. For example, in 2015, units with a well level were mainly distributed at the southwestern edge of the Hunan Province, southernmost point of the Jiangxi Province, south of the border of the Hunan and Jiangxi provinces, and on the border of the Hubei and Hunan provinces.

From the perspective of the time series, the number of research units at the well level decreased from eight in 2000 to four in 2005, and then continued to increase to 17 in 2015. The number of research units at the relatively well level continued to increase from 133 in 2000 and 2005 to 151 in 2015. The number of ordinary research units increased from 126 in 2000 to 131 in 2005, and then continued to decrease to 101 in 2015. However, the number of research units with weak and relatively weak levels insignificantly varied during the study period. The number of weak-level research units varied from 17 to 19 and those of relatively weak research units varied from 37 to 41. Therefore, the results of this study show that the ecosystem health in the MRYRUA remained relatively stable from 2000 to 2005. During the period 2005–2015, ecosystem health shifted from ordinary to well and relatively well levels. In 2015, the number of research units at the ordinary level, with Shaoyang as the center in the Hunan province and Nanchang as the center in the Jiangxi Province, significantly decreased compared with that in 2000, but the number of research units with weak and relatively weak levels barely changed.

### 3.2. Determination of Factors Controlling Ecosystem Health

Based on the GeoDetector simulation, the ecosystem health in the MRYRUA is affected by a variety of natural and socioeconomic factors (Figure 3). The effects of different factors on ecosystem health significantly differ. Based on the analysis of the q-statistic values at four time points, the controlling factors in 2000 can be ranked as follows: land use degree > proportion of forest land > population > proportion of construction land > DEM > precipitation > proportion of water. In 2005 and 2010, the ranking was as follows: land use degree > population > proportion of forest land > proportion of construction land > DEM > proportion of water > precipitation. The 2015 ranking is as follows: land use degree > proportion of construction land > population > proportion of forest land > DEM > proportion of water > precipitation. These results show that the land use degree has always been the main factor controlling the ecosystem health in the MRYRUA. The degree of land use reflects the degree of interference of the natural complex of land by socioeconomic factors. Therefore, it is an important influencing factor. However, the order of the other drivers changed. From 2000 to 2005, the population surpassed the proportion of forest land and became the second most important factor. The order of the factors did not change between 2005 and 2010. However, the rank of the proportion of construction land changed from fourth to second in 2015, which indicated that the ecosystem health in the MRYRUA was greatly affected by the proportion of construction land during this period, except for the degree of land use. This period was also characterized by the rapid construction of urban agglomerations in the MRYRUA, and the proportion of construction land rapidly increased.

### 3.3. Interactions among Drivers of Ecosystem Health

Based on the results of the interactive detection and analysis using the GeoDetector, the effects of the factors on ecosystem health were not independent from 2000 to 2015 but synergistic and mainly included bi- and nonlinear enhancements. However, the former was significantly stronger than the latter (Table 3). As the main factor, the degree of land use reflects the degree of interference in the development of the natural complex of land by human society. Therefore, the interactions between each factor and the degree of land use are the most complex. For example, only the interaction between the land use degree and average annual rainfall exhibited a nonlinear enhancement in 2005, 2010 and 2015. The interactions of the other factors with the land use degree were bi-enhanced. Further analysis revealed that the main interaction between the annual average precipitation and other factors was nonlinear enhancement.

## 4. Discussion and Implications

### 4.1. Rationality of the Ecosystem Health Assessment

Health, which was originally a medical concept that was mainly used to indicate the good state of the human body, was gradually used as a concept in animal and plant research [39]. The World Health Organization (WHO) define health as: “Health is a state of complete physical, mental, and social well-being and not only the absence of disease or infirmity”. Based on the development of the society, economy, and culture, the human living environment has also undergone various changes. The number of unfavorable changes in the living environment, such as global warming, extreme rainfall, increased atmospheric carbon dioxide concentration, soil erosion, salinization, reduction in wetlands, eutrophication of water bodies, loss of biodiversity, and depletion of self-heating resources, is much higher than that of favorable changes. These changes have become increasingly serious [40]. Therefore, the concept of health has shifted from organisms to the environment. Concepts, such as land, environmental, and ecosystem health, were proposed [41,42]. The emergence of these concepts provides an important theoretical basis for land use planning and ecosystem management.

Humans depend on a certain natural environment to survive. The quality of a region’s environment is directly related to human survival and reproduction. Humans should pay attention to the health of the living environment while maintaining their own health. Humans and their surroundings constitute today’s ecosystems. An ecosystem is a unified whole characterized by interactions among various organisms and between a biological community and its inorganic environment through energy flow and material circulation within a certain spatial and temporal range. In 1941, Aldo Leopold used soil as the research object and proposed a definition for land health, that is, the ability of land to renew itself. He used land sickness to describe soil erosion, fertility loss, hydrological anomalies, the invasion of species, and extinction of local species [41,43], which is also the original thought source of ecosystem health. Because of the deterioration of the global environment, ecosystems are severely damaged. Many scientists have started to study the health of ecosystems. Karr et al. showed that an ecosystem is considered to be healthy if it can realize its internal potential, has a stable state, can still repair itself when disturbed, and only requires minimal external support for management [44]. Schaeffer et al. discussed the measurement of ecosystem health based on human and animal health assessments and pointed out that methods and specific parameters and standards must be developed for the diagnosis of ecosystems and definition of ecosystem health, respectively [45]. In 1990 and 1991, seminars on ecosystem health were held in Maryland and Washington, USA, respectively. The purpose was to reach a consensus on the definition of ecosystem health. Finally, Haskell, Horton, and Costanza integrated the opinions of participating experts and scholars and defined ecosystem health as: “if an ecosystem is stable and sustainable, it is a healthy and free of disease symptoms system; in other words, the system can maintain its organizational structure in time, be able to self-adjust and have the ability to recover from coercion” [42,46]. Consequently, the initial concept of ecosystem health took shape.

At present, ecosystems on Earth are generally affected by humans [47], and global ecosystems are generally degraded. Therefore, the improvement of ecosystem health assessment methods is an urgent task [48,49]. Whether it is for individuals, populations, or ecosystems, health evaluation is somewhat subjective because the definition of the “health” state is affected by the values and understanding of human society. The ecosystem health standard is a human standard [50]. Health and sustainability are both scale-dependent concepts [51]. Therefore, different quantitative evaluation methods at different scales must be established. The evaluation methods of ecosystem health mainly include indicator species and index system methods. The indicator species method is used to indirectly assess the health status of an ecosystem by monitoring the response of the indicator species in the ecosystem to environmental stress, such as the population size, biomass, age structure, toxicological response, diversity, and important physiological indicators. This method is relatively simple and easy to operate, but it lacks indicator species screening criteria. The quality of indicator species selection directly affects the reliability of the evaluation results. Because of the complexity of urban ecosystems, this method is not suitable. As a comprehensive analysis method, the index system method combines physical, chemical, and biological methods and is based on conventional methods utilized in botany, soil science, ecology, physiology, and toxicology to synthesize a large amount of complex information. Index system methods are currently the most commonly used methods worldwide. The index system can be an index system composed of pure natural indicators or a composite index system consisting of multiple indicators of nature, society, and economy. The methodological assessment of ecosystem health is advantageous for the comprehensive understanding of the health of all aspects of the structure and function of an ecosystem.

Ecosystem health assessment is generally very complex because each ecosystem has different basic characteristics [52], and the structure and function of the ecosystem may differ at different spatiotemporal scales. Therefore, many model evaluation methods were developed. Each model has different advantages and disadvantages and is suitable for different research fields. The PSR and VOR (the ecosystem vigor, organization, resilience) are the two most commonly used methods for ecosystem health assessment. The PSR focuses on the state of ecosystems and the effects of human activities on ecosystems and their interactions [53]. It is suitable for large- and medium-sized regional ecosystems that are greatly affected by human activities. It focuses not only on the characteristics of ecosystem health, but also on the interactions between natural characteristics and human attributes [54]. However, VOR pays more attention to the health of an ecosystem’s structure [42]. The model assesses ecosystem health based on the ecosystem vitality, organization, and resilience. However, based on the above-mentioned models, ecosystem health is generally assessed by considering the state of the ecosystem itself and external disturbances, but the ecosystem’s ability to provide services to humans is ignored [15,55]. Urban ecosystem health assessments must meet the requirements of human beings and maintain their own sustainability [15]. Ecosystem services reflect the direct connection between the human society and ecosystem, respond to various ecological factors and changes in the social environment, and reflect the value and significance of human survival and development. Therefore, it is very important to add ecosystem services to an evaluation index system [18]. Scholars developed a VORS model based on the VOR model [15,56]. It considers the services provided by the ecosystem, that is, the direct and indirect ecological functions it provides to humans [57]. The VORS model combines the structure of the ecosystem and its ecosystem services for humans [15]. It is easy to describe in theory and practice [52]; therefore, it is suitable for the health assessments of urban ecosystems with strong human interference. Based on the VORS, an ecosystem health assessment system was built for the MRYRUA in this study, providing a more comprehensive and accurate evaluation of the urban ecosystem health. In addition, work at the county level was based on second-level land types such that the research results better reflected regional, ecological, environmental construction.

### 4.2. Analysis of Mechanisms Underlying Factors Affecting Ecosystem Health

We evaluated the spatial distribution of ecosystem health in the MRYRUA over four years. The results show that weak and relatively weak ecosystem health is mainly distributed in the central areas of urban groups. These areas generally have a high population density, economic development, and proportion of construction land. This is consistent with the results of previous studies on the ecosystem health of urban agglomerations. For example, lower ecosystem health values were reported for areas with faster population and economic development in the Shanghai–Hangzhou Bay Metropolitan Area [58]. This is also similar to the results of studies of individual, large cities such as Shenzhen [15], Beijing [7], and Guangzhou [59]. On the one hand, the increase in the proportion of construction land for urbanization requires a large amount of land resources; changes the physical soil characteristics; and reduces drainage, heat absorption, and heat dissipation, which ultimately affects the function of the soil system [60]. On the other hand, the effect of the increase in urban construction land on the ecosystem is mainly due to an increase in the population density. Because of the reduction in species populations and the decline of ecosystem functions, the discharge of garbage and sewage often leads to an unbalanced ecosystem [61]. The results of our study show that the degree of land use is the main factor affecting ecosystem health. The development, utilization, and protection of land resources are some of the major social issues related to China’s rapid urbanization process. The land use degree is a comprehensive indicator that directly affects the health of the ecosystem [62,63].

Although the research units with relatively weak and weak levels in the central areas of the three major metropolitan regions insignificantly changed during the study period, the units with an ordinary level in the surrounding areas gradually transformed to units with relatively well and well levels, especially after 2005. Therefore, the overall ecosystem health of the MRYRUA is improving. This may be due to the fact that the Hubei Wuhan Metropolis and Hunan Changsha–Zhuzhou–Xiangtan City Group were approved to become the national “two-type society” construction comprehensive reform pilot zone in 2006. Jiangxi’s Poyang Lake Ecological Economic Zone Plan was approved by the State Council. The MRYRUA has entered a new stage of development. In 2010, the “Middle Yangtze River Region” composed of the Wuhan Metropolis, Changsha–Zhuzhou–Xiangtan City Group, and Poyang Lake Ecological Economic Zone was listed as “National Key Development Area”. In 2012, “Several Opinions of the State Council on Vigorously Implementing the Strategy for Promoting the Rise of the Central Region” clearly stated: “Encourage and support the Wuhan Metropolis, Changsha–Zhuzhou–Xiangtan City Group, and Poyang Lake City Group to carry out strategic cooperation to promote the MRYRUA integrated development”. The implementation of these policies will promote the healthy development of urban agglomerations in the MRYRUA. In recent years, the development of the MRYRUA has become important for China’s economic development, and a series of planning and development plans were introduced. For example, the National Development and Reform Commission formally issued the “Thirteenth Five-Year Plan for Promoting the Rise of the Central Region” in 2016. The “Opinions of the Central Committee of the Communist Party of China on Establishing a New Effective Mechanism for Regional Coordinated Development” require Wuhan to be the center of development in the MRYRUA in 2018. In the future, the MRYRUA will develop in a direction of more economic progress and a more harmonious environment.

### 4.3. Policy Implications

Determining regional differences in the ecosystem health and its driving factors in the MRYRUA is important for scientific research, as well as for policy design and regional sustainable development [55,64,65]. The results of this study provide important suggestions for the formulation of ecological protection and restoration measures in the MRYRUA. Our results are consistent with those of previous studies on urban ecosystem health. Lower ecosystem health values are often observed in areas with large populations and in economically developed areas [15,58]. This shows that more attention was paid to the growth of the GDP (Gross Domestic Product) in the development of regions in recent decades and the protection of ecosystem health was ignored. Although the ecosystem health of the MRYRUA is improving, the ecosystem health in the main urban centers remains grim. Therefore, attention should be paid to the ecosystem health in the central area, while enhancing the ecosystem health of big cities. The degree of land use and proportion of construction land are important factors that will affect the urban ecosystem health in the future. Therefore, the rationality of the transformation of land use types should be emphasized during the construction of urban agglomerations. In addition, natural ecosystems should be protected and different ecological protection measures based on the health of regional ecosystems should be implemented. Priority should be given to areas with low ecosystem health levels, urban green spaces should be increased, and ecological red lines should be set. In the central areas of Wuhan Metropolis, Changsha–Zhuzhou–Xiangtan City Group, and Poyang Lake City Group, the effects of land use and the proportion of construction land should be reduced, and the proportion of urban green spaces and ecological protection should be increased.

### 4.4. Limitations and Future Directions

In this study, we used the VORS framework and a GeoDetector to evaluate ecosystem health and analyze its driving factors in the MRYRUA, respectively. Our results provide guidance for the assessment and development of ecosystem health in this region. However, this study also has certain limitations: (1) Although the VORS framework can be applied to all types of ecosystems, the increase in the ecosystem service dimension places an emphasis on human welfare. The quantification of the indicators of ecosystem organization, resilience, and services depends on land use data. It must be determined how more data sources can be incorporated. Processes that can be used to characterize ecosystem organization, resilience, and services differ. The effects of the spatial adjacency of land use types should be considered [15]. Although human well-being depends on ecosystem services, different types of ecosystem services correspond to different human needs and have different effects on well-being; (2) The GeoDetector is an important statistical method for detecting spatial differentiation and revealing the factors controlling it. In this study, relatively few control factors—population, land use, and geoclimatic factors—were considered, but there is a lack of economic factors. Therefore, the role of economic factors should be included in future studies to better assess ecosystem health.

## 5. Conclusions

In this study, the ecosystem health in the MRYRUA in 2000, 2005, 2010 and 2015 was evaluated based on four dimensions: ecosystem vitality, organization, resilience and services. A GeoDetector was used to quantitatively analyze the effects of seven factors: the proportion of construction land, proportion of forest land, proportion of water, land use degree, population, average annual precipitation and DEM, on ecosystem health in different periods. From our results, we concluded that: (1) there are significant spatial differences in ecosystem health in the MRYRUA. The central areas of the Wuhan Metropolis, Changsha–Zhuzhou–Xiangtan City Group and Poyang Lake City Group spread out as the center and the ecosystem health level continues to improve. (2) Temporally, the number of units with a well and relatively well level increased, that of units with an ordinary level gradually decreased, and that of units with weak and relatively weak levels remained stable. The average ecosystem health in the MRYRUA is improving. (3) Ecosystem health in the MRYRUA is mainly affected by the degree of land use. Bi-enhanced or nonlinear interactions among the different control factors were observed. (4) In 2015, the effect of the proportion of construction land on ecosystem health increased, ranking second. The results of this study provide guidance for the establishment of land use norms in the MRYRUA. The layout of the urban agglomeration during the construction of different land types significantly affects ecosystem health.

## Figures and Tables

**Figure 1 ijerph-19-00771-f001:**
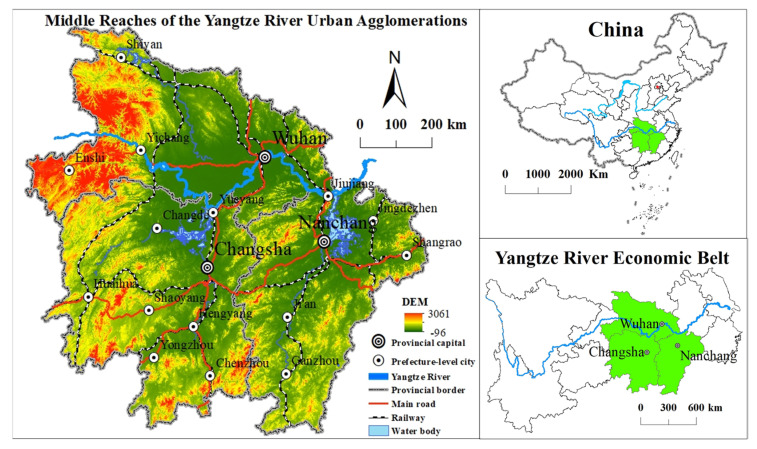
Location of the MRYRUA in China.

**Figure 2 ijerph-19-00771-f002:**
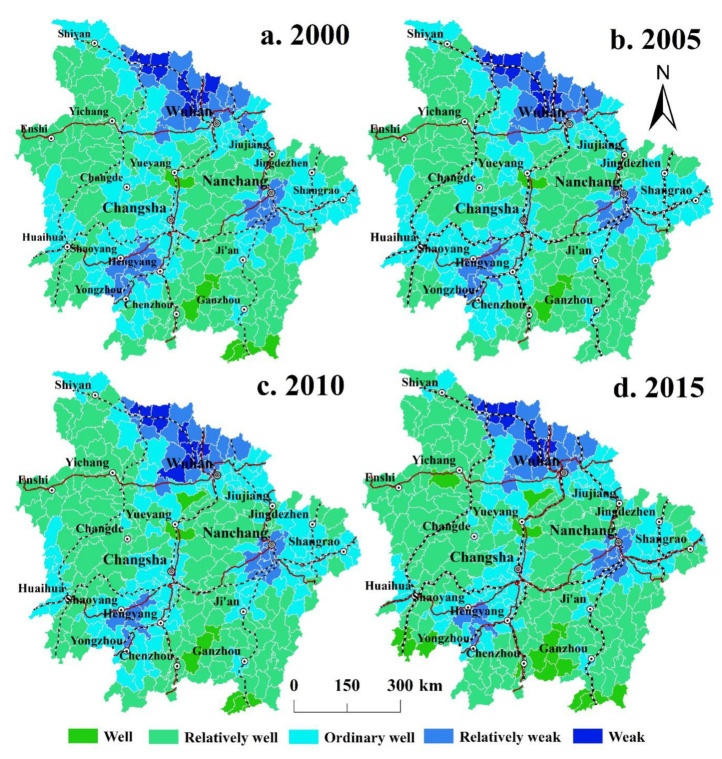
Ecosystem health levels in the MRYRUA from 2000 to 2015.

**Figure 3 ijerph-19-00771-f003:**
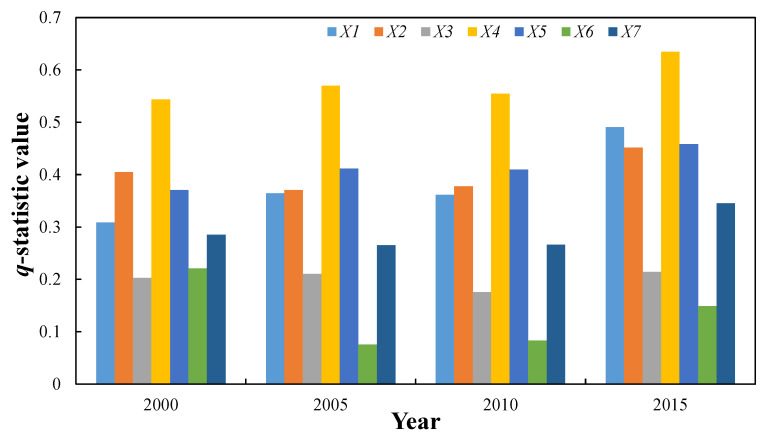
Contributions of controlling factors from 2000 to 2015.

**Table 1 ijerph-19-00771-t001:** Land use classification and meaning, ecosystem resilience (RC), and services coefficient (SC).

First-Level Land Types		Second-Level Land Types			RC	SC
No.	Name	No.	Name	Meaning		
1	Cultivated field	11	Paddy fields	Refers to the arable land with guaranteed water sources and irrigation facilities, which can be normally irrigated in normal years for the cultivation of aquatic crops, such as rice and lotus roots, including arable land with rice and dry land crop rotation.	0.35	3.89
		12	Dry land	Refers to the arable land without irrigation water sources and facilities, which relies on natural water to grow crops; dry crop arable land with water sources and irrigation facilities, which can be normally irrigated in a normal year; arable land dominated by vegetable cultivation.	0.3	4.01
2	Forest land	21	Woodland	Refers to natural forests and plantations with a canopy closure > 30%, including timber forests, economic forests, shelterbelts, and other forest plots.	0.85	22.95
		22	Shrub forest	Refers to low woodland and shrubland with a canopy density > 40% and height below 2 m.	0.80	15.22
		23	Sparse forest land	Refers to forest land with a canopy density ranging from 10–30%.	0.75	15.16
		24	Other forest land	Refers to unforested afforestation sites, ruins, nurseries, and various types of gardens (e.g., orchards, mulberry gardens, tea gardens, hot plantation forest gardens)	0.60	14.12
3	Grassland	31	High-cover grassland	Refers to natural grassland, and improved and cut grassland with a cover > 50%. Such grasslands are generally characterized by better water conditions and dense grass cover.	0.50	5.62
		32	Medium-cover grassland	Refers to natural and improved grasslands with a cover of 20–50%. Such grasslands generally have insufficient water sources and sparse grass cover.	0.45	5.07
		33	Low-cover grassland	Refers to natural grassland with a cover of 5–20%. This type of grassland lacks water, the grass is sparse, and the conditions for pastoral use are poor.	0.40	5.07
4	Waters	41	River canal	Refers to naturally formed or artificially excavated rivers and land below the main trunk perennial water level. Artificial canals include embankments.	0.85	125.61
		42	Lake	Refers to the land below the perennial water level in a naturally formed water accumulation area.	0.85	125.61
		43	Reservoir pit	Refers to the land below the perennial water level in artificially constructed water storage areas.	0.80	125.61
5	Urban and rural, industrial and mining, residential land	51	Urban land	Refers to land in large, medium, and small cities and built-up areas above county towns.	0.20	0
		52	Rural settlement	Refers to rural settlements independent of towns.	0.25	0
		53	Other construction land	Refers to sites such as factories and mines, large industrial areas, oil fields, salt fields, and quarries, as well as transportation roads, airports and special sites.	0.15	0
6	Unused land	61	Bare land	Refers to the land covered by surface soil; the vegetation cover is below 5%.	0.95	0.20
		62	Bare rock texture	Refers to the surface of rock or gravel, covering more than 5% of the land.	0.95	0.20
7	Wetlands	71	Beach	Refers to the land between the water level of rivers and lakes in the normal water period and the water level in the flood season.	0.70	52.02
		72	Marsh land	Refers to flat and low-lying land, poor drainage, long-term humidity, seasonal water accumulation or perennial water accumulation, and the growth of wet plants on the surface.	0.70	52.02

**Table 2 ijerph-19-00771-t002:** Index system of urban ecosystem health assessment based on the VORS framework.

Target Layer	Criterion Layer	Index Layer	Explanation
Ecosystem health	Ecosystem vigor	NPP	The total amount of net organic matter produced by photosynthesis. The greater the NPP is, the more vital is the ecosystem.
	Ecosystem organization	SHDI	The higher the SHDI is, the higher the heterogeneity and the stronger the organization of the landscape.
		CONTAG	A high CONTAG indicates that a certain dominant patch in the landscape has formed a good connectivity, that is, the higher the spread is, the better the connectivity and the stronger the organization of the landscape.
		Landscape fragmentation index (LFI)	Refers to the degree of fragmentation of the landscape and reflects the overall spatial complexity of the landscape in the study area. The value ranges between 0 and 1. The closer the FNI (fragmentation index) is to 1, the greater the degree of landscape fragmentation.
		COHESION	A high COHESION indicates that the patch type has a higher degree of aggregation in the landscape, that is, the higher the patch COHESION is, the better the connectivity and the stronger the organization of the landscape.
		AWMPFD	It is an important indicator that reflects the overall characteristics of the landscape pattern. It also reflects the effects of human activities on the landscape pattern. The value ranges between 1 and 2. The value of natural landscapes that are less affected by human activities is high, whereas the value of artificial landscapes that are greatly affected by human activities is low.
	Ecosystem resilience	RC	The resilience coefficient is set according to the difficulty with respect to the recovery of different land use types. The value ranges between 0 and 1.
	Ecosystem services	SC	Refer to Xie et al. for Chinese terrestrial ecosystem services coefficients (SCs) and set the SCs of land use types [32].

**Table 3 ijerph-19-00771-t003:** Interaction detectors of the GeoDetector.

	X1	X2	X3	X4	X5	X6	X7
2000							
X1	0.308						
X2	0.594□	0.405					
X3	0.508□	0.649Δ	0.202				
X4	0.603□	0.609□	0.698□	0.544			
X5	0.424□	0.530□	0.572□	0.609□	0.371		
X6	0.550Δ	0.689Δ	0.541Δ	0.730□	0.599Δ	0.221	
X7	0.498□	0.509□	0.417□	0.649□	0.505□	0.640Δ	0.286
2005							
X1	0.365						
X2	0.570□	0.371					
X3	0.538□	0.660Δ	0.211				
X4	0.600□	0.640□	0.753□	0.570			
X5	0.484□	0.542□	0.615□	0.614□	0.412		
X6	0.473Δ	0.539Δ	0.481Δ	0.664Δ	0.534Δ	0.075	
X7	0.519□	0.470□	0.459□	0.668□	0.542□	0.542Δ	0.265
2010							
X1	0.362						
X2	0.595□	0.378					
X3	0.552Δ	0.666Δ	0.176				
X4	0.633□	0.615□	0.729□	0.555			
X5	0.494□	0.532□	0.579□	0.610□	0.410		
X6	0.574Δ	0.608Δ	0.445Δ	0.701Δ	0.618Δ	0.083	
X7	0.492□	0.457□	0.450Δ	0.627□	0.532□	0.527Δ	0.266
2015							
X1	0.491						
X2	0.669□	0.452					
X3	0.618□	0.675Δ	0.214				
X4	0.681□	0.672□	0.750□	0.634			
X5	0.557□	0.604□	0.623□	0.657□	0.458		
X6	0.683Δ	0.661Δ	0.468Δ	0.786□	0.678Δ	0.149	
X7	0.601□	0.519□	0.487□	0.696□	0.588□	0.532Δ	0.345

Notes: (Δ) denotes the nonlinear enhancement of two variables and (□) denotes the bi-enhancement of two variables.

## Data Availability

National Science and Technology Basic Condition Platform National Earth System Science Data Center (http://www.geodata.cn); Geospatial Data Cloud (http://www.gscloud.cn/); Resource and Environmental Science Data Center of the Chinese Academy of Sciences (http://www.resdc.cn/Default.aspx); Worldpop (https://www.worldpop.org/). Details in 2.2 data sources and processing.

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
