# Peer review of "Assessment of Ecosystem Health and Its Key Determinants in the Middle Reaches of the Yangtze River Urban Agglomeration, China"

_ijerph, 2022, doi:10.3390/ijerph19020771_

Round 1
Reviewer 1 Report
General overview
The authors assessed the ecosystem health in the middle reaches of the Yangtze River Urban Agglomeration (MRYRUA) using the ecosystem vigour, organization, resilience, and services framework at the county scale. The MRYRUA is a nationwide large-scale urban agglomeration that includes the Wuhan metropolis, the Changsha-Zhuzhou-Xiangtan city group and the Poyang Lake city group, which play an important role in China's regional development model. Findings showed that there are significant spatial differences in ecosystem health in the MRYRUA, and the layout of the urban agglomeration during the construction of different land types significantly affects the ecosystem health.
The authors addressed a very interesting topic.
The methods used are correct. The study design is sound. The statistical analyses are correct and adequate for the purposes of the study. The discussions correctly describe the results obtained and how the study supports or criticizes current knowledge. Limitations of the study are described.
Specific comments
I have some suggestions for improving the manuscript.
-The introduction is too long (about 1300 words). It must be more concise and clearer.
It would be ideal to have an introduction of a maximum of 600/700 words (known information (about 1 paragraph), unknown information (about 1 paragraph), your burning question/hypothesis/aim, and your experimental approach to fill the gaps (about 1 paragraph). Detailed descriptions, speculations, and criticisms of studies should be included in the discussions.
-Finally, you must use keywords other than those present in the title to optimize the search for the manuscript through search engines. Replace “Ecosystem health, Yangtze River Urban Agglomeration and China " with other appropriate keywords.
Author Response
Thank you very much for your suggestions. I have made the modifications according to the two suggestions you raised, and the explanations are as follows:
First, I delete the concept and origin of “health” ,because I think that it is unnecessary in the introduction to write so much about the general literature. Much of the information here is already written at the start of the discussion section anyway. In addition I also delete the explanation of “Stress”, “State”, “Response”, because this appears to be an unnecessarily long description of this model, which could be removed without negatively affecting my paper. Moreover, I also reworked the statement to make it more concise.
Second, I use the keywords of “Spatiotemporal pattern”; “GeoDetector”; “sustainable development”; “ecological environment” to replace “Ecosystem health, Yangtze River Urban Agglomeration and China ".
Thank you very much again for your efforts and comments.

Reviewer 2 Report
The issues related to the ecosystem’s health should constitute an important area of scientific analysis nowadays. In this context, I consider the presented paper to be a very important voice in the discussion on this topic. I believe that the paper should be published after taking into account the following comments:
Abstract – prepared in accordance with current standards
The Introduction section needs some more information. It should state the purpose of the work in the form of the research problem supported by a hypothesis or a set of questions, explaining briefly the methodological approach used to examine the research problem, highlighting the potential outcomes your study can reveal, and outlining the remaining structure and organization of the paper.
Materials and Methods section: Considering the number and type of variables used to develop the model, it should be concluded that the assessment of ecosystem health and its key determinants in the middle reaches of the Yangtze River Urban Agglomeration is comprehensive. It is only a pity that the latest data is only for 2015, but I understand that not everything depends on the authors of the study.
Results section: no comments to this part.
Discussion and implications: Authors have included all the elements required for a discussion section. They interpreted and described the significance of their findings in light of what was already known about the research problem being investigated and to explained any new understanding or insights that emerged as a result of their study of the problem.
In Conclusions section I need stronger explanation why the research should matter to the readers after they have finished reading the paper.
Summing up, I believe that the presented article meets the requirements for scientific papers. It should be published after taking into account the above comments.
Author Response
Thank you very much for your suggestions. I have made the modifications according to the four suggestions you raised, and the explanations are as follows:
First, I have I have modified it according to current standards, included that the statement was remodified and the key words was modified.
Second, your comments on the introduction part are very pertinent. I modified part of the content to make the article clearer, but I also deleted part of the irrelevant content. Thank you for your understanding. The modifications are as follows: I delete the concept and origin of “health” ,because I think that it is unnecessary in the introduction to write so much about the general literature. Much of the information here is already written at the start of the discussion section anyway. In addition I also delete the explanation of “Stress”, “State”, “Response”, because this appears to be an unnecessarily long description of this model, which could be removed without negatively affecting my paper. Moreover, I also reworked the statement to make it more concise.
Third, the update time of public data of different factors is inconsistent, some in 2015, some in 2018, and some in 2019. Considering that the article is calculated according to a period of five years, I can only choose the data of 2015. If there are updated data later, I will continue to study, update my results and share them with you. Thank you very much for your understanding.
Forth, The main research content of this paper is ecosystem health and its influencing factors. The correlation between different factors and ecosystem health is the main conclusion, which can guide the government's future ecological protection, and I think the necessity of its research has been clearly explained.
At last, I hope my answer can make you satisfied, thank you for your efforts!

Reviewer 3 Report
Paper needs more work.; The intro doesn't flow and is hard to follow ;
The methods are not clear
Author Response
Thank you very much for your suggestions. I have made the modifications according to the three suggestions you raised, and the explanations are as follows:
First, I have done some work again, mainly including the introduction of the modification, modification of sentences, so that the article read more interesting. The modifications are as follows: I delete the concept and origin of “health” ,because I think that it is unnecessary in the introduction to write so much about the general literature. Much of the information here is already written at the start of the discussion section anyway; In addition I also delete the explanation of “Stress”, “State”, “Response”, because this appears to be an unnecessarily long description of this model, which could be removed without negatively affecting my paper. Moreover, I also reworked the statement to make it more concise; I use the keywords of “Spatiotemporal pattern”; “GeoDetector”; “sustainable development”; “ecological environment” to replace “Ecosystem health, Yangtze River Urban Agglomeration and China ". Moreover, I also reworked the statement to make it more concise.
Second, I feel very sorry for your hard to follow because of grammar. I have made a lot of modifications,for example, in first sentence of “2.1 Study area” This seemed like an overly general statement. Any region can be said to extend in the four directions. You could think that it stretched into the four regions of China: West, East, North, and South China, but according to Figure 1, this does not appear to be the case given that the western boundary of the MRYRUA appears to be Hubei and Hunan Provinces in Central and South-Central China. I have revised the sentence for clarity. And I will not list the other modifications one by one. I am sorry again for the inconvenience of your reading, thank you very much for your work and advice.
Third, The method is not clear because I introduced some other methods. I deleted some irrelevant methods and further described the method used in the study to make it more clear.
I attach great importance to your opinions on modification, and I have modified them one by one. The modification is mentioned above and the modification content is in the text. Finally, thank you very much for your work and advice, wish you a happy life!

Round 2
Reviewer 1 Report
Thanks for your replies!
Good luck for your future projects.